# Polyaniline/Ag₂S–CdS Nanocomposites as Efficient Electrocatalysts for Triiodide Reduction in Dye-Sensitized Solar Cells

**Meng Kuo [1], Tsung-Chia Cheng [1], Huai-Kai Ye [1], Tzong-Liu Wang [2], Tzu-Ho Wu [3], Chi-Ching Kuo [4] and Rong-Ho Lee [1,\*]**

[1] Department of Chemical Engineering, National Chung Hsing University, Taichung 402, Taiwan; ab8654029@gmail.com (M.K.); g108065004@mail.nchu.edu.tw (T.-C.C.); kyleleaf99@gmail.com (H.-K.Y.)

[2] Department of Chemical and Materials Engineering, National University of Kaohsiung, Kaohsiung 811, Taiwan; tlwang@nuk.edu.tw

[3] Department of Chemical and Materials Engineering, National Yunlin University of Science and Technology, Yunlin 64002, Taiwan; wutzu@yuntech.edu.tw

[4] Institute of Organic and Polymeric Materials, Research and Development Center of Smart Textile Technology, National Taipei University of Technology, Taipei 10608, Taiwan; kuocc@mail.ntut.edu.tw

\* Correspondence: rhl@nchu.edu.tw

**Abstract:** In this study, an Ag₂S–CdS nanocomposite (AC11) was prepared through chemical co-precipitation of silver nitrate and cadmium acetate in an aqueous solution of thiourea. We then synthesized PACI, a nanocomposite of polyaniline (PANI) and AC11, through in situ polymerization of aniline in an AC11-containing solution, resulting in uniform embedding of the AC11 nanoparticles in the PANI fibers. Moreover, we synthesized the nanocomposite PACO through deposition of the AC11 nanoparticles on the surface of the PANI fibers. PANI, PACI, and PACO were then spin-coated onto conducting glasses to form PANI-S, PACI-S, and PACO-S counter electrodes, respectively, for dye-sensitized solar cells (DSSCs). Cyclic voltammetry revealed that the electrochemical catalytic activity of the PACI-S electrode was much higher than those of the PANI-S and PACO-S electrodes. Furthermore, the photovoltaic properties of the PACI-S-based DSSC were much better than those of the PANI-S- and PACO-S-based DSSCs. Indeed, the highest short-circuit current density (12.06 mA/cm²), open-circuit voltage (0.72 V), fill factor (0.58), and photoenergy conversion efficiency (5.04%) were those of the DSSC featuring PACI-S as the counter electrode.

**Keywords:** polyaniline; Ag₂S–CdS; electrochemical catalytic activity; dye-sensitized solar cell

## 1. Introduction

Dye-sensitized solar cells (DSSCs) have potential for application in power generation because of their low cost, flexibility, and efficient photovoltaic (PV) conversion of solar energy [1,2]. A DSSC typically comprises a dye-sensitized TiO₂ photoactive anode, an iodide/triiodide redox electrolyte, and a platinum (Pt) counter electrode [3,4]. The Pt electrode is employed to reduce I₃⁻ to I⁻ to complete the cycle of electron transfer in the DSSC [1,2]. Unfortunately, the high price of Pt film limits the commercial applications of DSSCs. Moreover, Pt-based counter electrodes generally undergo corrosion mediated by the iodine liquid electrolyte, resulting in DSSCs having low stability [5]. Therefore, many alternative materials have been studied as counter electrodes, including conducting polymers [6,7], carbon-based nanocomposites [8–12], and transition metal oxides, nitrides, carbide, selenides, and sulfides [13–21].

Conducting polymers are promising materials for use in counter electrodes because of their inexpensiveness, high conductivity, and excellent catalytic activity for the reduction of I₃⁻ [22–24]. In particular, polyaniline (PANI) has been used as a counter electrode

because of its ease of synthesis, high conductivity, and excellent electrocatalytic activity [22]. In addition, quasi-solid-state DSSCs have been fabricated featuring a polypyrrole (PPy)-functionalized counter electrode [23]. A thin film of PPy was fabricated on a substrate coated with fluorine-doped tin oxide (FTO) through simple electrodeposition from an aqueous solution using pyrrole as the precursor monomer. The photoenergy conversion efficiency (PCE) of the PPy-based device was approximately 21% lower than that of the reference Pt-based device [23]. A high-conductivity thin film of poly(3,4-ethylenedioxythiophene) (PEDOT) has also been coated on FTO glass for use as a counter electrode in a DSSC [24]. This PEDOT-based counter electrode displayed outstanding electrocatalytic activity for the $I_3^-/I^-$ reduction, as well as a charge-transfer resistance lower than that of the Pt electrode. Nevertheless, the PV performance of DSSCs fabricated with counter electrodes based on conducting polymers has not been as high as that of Pt-based DSSCs [24].

Transition metal sulfides are also promising counter electrode materials because of their cheap feedstocks, high conductivity, and excellent catalytic activity [25]. Wang et al. reported that cobalt sulfide (CoS) is an effective catalyst for the reduction of $I_3^-$ to $I^-$ in a DSSC, with PV properties exceeding those of Pt as an electrocatalyst [26]. Molybdenum sulfide ($MoS_2$) and tungsten sulfide ($WS_2$) have also been tested as counter electrode catalysts in DSSCs [27]. Both $MoS_2$ and $WS_2$ display high electrocatalytic activities for the regeneration of the redox couple ($I_3^-/I^-$). The $MoS_2$- and $WS_2$-based DSSCs achieved high PCEs (7.59 and 7.73%, respectively) and PV properties close to those of Pt-based DSSCs. Sun et al. reported the use of an electrochemically deposited nickel sulfide (NiS) film as a counter electrode displaying high electrocatalytic activity for the reduction of $I_3^-$ to $I^-$ [28]; the PCE of their NiS film-based DSSC was comparable to that of the corresponding cell featuring a Pt-based counter electrode. Several other metal sulfides—lead sulfide (PbS), silver sulfide ($Ag_2S$), copper monosulfide (CuS), cadmium sulfide (CdS), and zinc sulfide (ZnS)—have also been tested as counter electrodes as replacements for the expensive Pt in DSSCs [29]. Relative to a Pt-based DSSC, higher PCEs were obtained for the corresponding DSSCs fabricated with PbS-, $Ag_2S$-, and CuS-based counter electrodes, resulting from the improved FFs arising from their excellent electrocatalytic activities.

Although high catalytic activity has been observed for counter electrodes based on transition metals, a lack of connectivity between nanoparticles (NPs) of transition metal materials would result in the fewer pathways for electron transfer in such counter electrodes. Moreover, NP-type transition metal materials usually adhere poorly to indium tin oxide (ITO)- or FTO-deposited conducting substrates. Recently, transition metal material/conducting polymer nanocomposites displaying large electrocatalytic activities and high conductivities have been developed for use as the counter electrodes of DSSCs [30,31]. The conducting polymer PEDOT has functioned as a good binder for NPs of silicon carbide (SiC-NPs), improving the contact between the SiC-NPs and promoting electron transfer in a counter electrode based on the SiC-NP/PEDOT composite [31]. Zatirostami et al. reported the PV properties of a DSSC incorporating a counter electrode based on a $WO_3$/PANI nanocomposite [32]. The PCE of the $WO_3$/PANI-based DSSC was better than that of the Pt-based DSSC, arising from the high electrical conductivity and suitable electrocatalytic behavior of the produced material. In addition, $MoS_2$/PEDOT nanocomposites have been prepared as counter electrodes, taking advantage of the high electrocatalytic ability and conductivity of both $MoS_2$ and PEDOT [33]; compared with corresponding PEDOT- and Pt-based DSSCs, the PCE of the $MoS_2$/PEDOT-based DSSC was improved by 10.6 and 6.4%, respectively.

$Ag_2S$–CdS nanomaterials have been synthesized for their use in various applications, including antimicrobial agents, photocatalysts, energy conversion materials, and organic–inorganic hybrid solar cells [34–39]. Xu et al. reported that heterojunctions based on CdS–$Ag_2S$ displayed higher light absorption intensities and stronger photocurrent responses than those of a thin film of CdS; hybrid solar cells fabricated with FTO/CdS/$Ag_2S$/poly(3-hexylthiophene) (P3HT)/Au provided a PCE higher than that of the corresponding

FTO/CdS/P3HT/Au cell [39]. In this present study, we prepared an Ag$_2$S–CdS nanocomposite (AC11) through the chemical co-precipitation of silver nitrate and cadmium acetate in thiourea aqueous solution. Moreover, we prepared the nanocomposite PACI through in situ polymerization of aniline in a solution containing AC11 and the nanocomposite PACO through co-precipitation of AC11 NPs on the surface of PANI fibers. We then fabricated PANI-, PACI-, and PACO-coated FTO glasses through spin-coating to function as PANI-S, PACI-S, and PACO-S counter electrodes, respectively, of DSSCs. Scanning electron microscopy (SEM) and transmission electron microscopy (TEM) revealed the morphologies of the PANI-S, PACI-S, and PACO-S counter electrodes; cyclic voltammetry (CV) revealed their electrochemical catalytic activities. Herein, we also describe the PV performance of DSSCs fabricated with the PANI-S, PACI-S, and PACO-S counter electrodes.

## 2. Results and Discussion

### 2.1. Characterization of Ag$_2$S–CdS Nanocomposite

Figure 1 presents SEM images of the Ag$_2$S, CdS, and AC11 samples. The Ag$_2$S particles aggregated to form a porous network structure with many open pores and channels (Figure 1a); the average diameter of the Ag$_2$S particles ranged from 0.3 to 1.5 μm. In contrast, the CdS particles were smaller, ranging in size from 0.1 to 0.5 μm (Figure 1b). Figure 1c,d reveal that the particles of the Ag$_2$S–CdS nanocomposite AC11 (100–500 nm) were much smaller than those of the Ag$_2$S and CdS samples, presumably because addition of the surfactant (SDS) in the reaction mixture inhibited aggregation of the AC11 NPs.

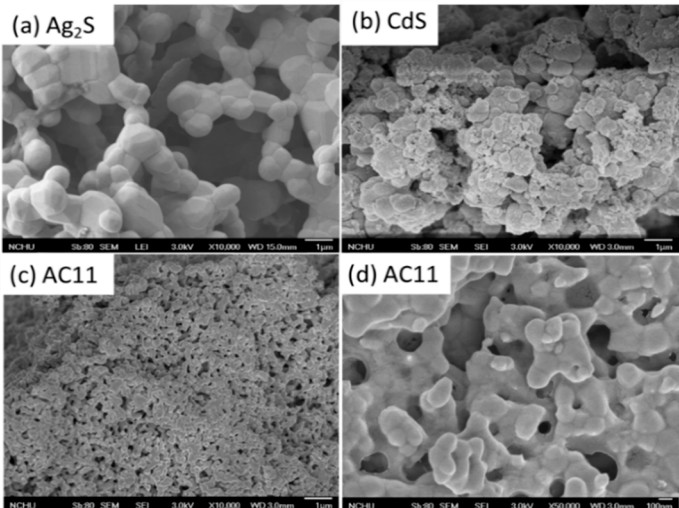

**Figure 1.** SEM images of the samples of (**a**) Ag$_2$S, (**b**) CdS, and (**c,d**) AC11.

Figure 2 displays the XRD patterns of the Ag$_2$S, CdS, and AC11 samples. For the Ag$_2$S sample, peaks appeared at values of $2\theta$ of 26.3, 29.1, 31.4, 34.4, 36.8, 37.7, 40.7, 43.4, 46.2, 47.8, 48.8, 53.8, 58.4, and 63.8°, representing the (101), (110), (−112), (120), (112), (−103), (031), (103), (123), (113), (−212), (−213), (−223), and (−134) planes, respectively, of the JCPDS14-0072 structure [40,41]. For the CdS sample, diffraction peaks appeared at values of $2\theta$ of 26.60, 44.17, and 52.46°, representing the (111), (220), and (311) planes, respectively, of the JCPDS 10-454 structure [41]. The XRD pattern of the Ag$_2$S–CdS nanocomposite AC11 featured diffraction peaks arising from Ag$_2$S that were of much greater intensity than those arising from CdS.

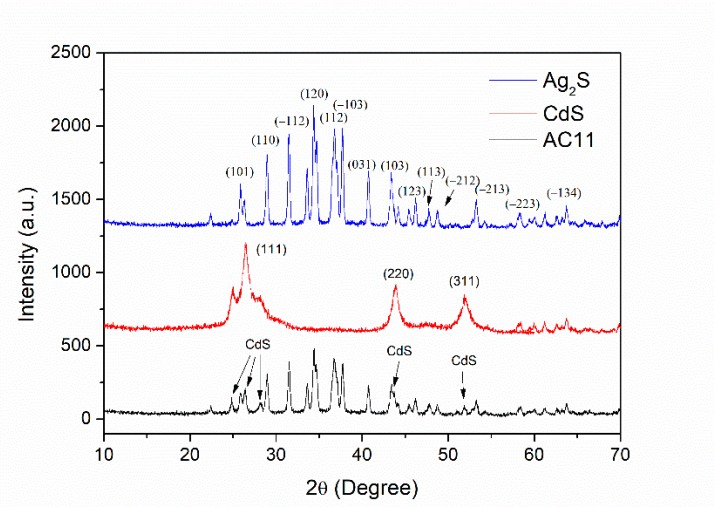

**Figure 2.** XRD patterns of the samples of Ag₂S, CdS, and AC11.

Figure 3 provides XPS spectra of the Ag₂S–CdS nanocomposite AC11. The Ag 3d spectrum of AC11 featured two peaks at 374.0 and 368.0 eV (Figure 3a); we assign them to Ag $3d_{3/2}$ and Ag $3d_{5/2}$, respectively [42]. Moreover, the Cd 3d spectrum (Figure 3b) contained peaks at 412.0 and 405.1 eV that are characteristic of the binding energies of Cd $3d_{3/2}$ and Cd $3d_{5/2}$, respectively [43]. The two deconvoluted peaks in the S 2p spectrum at 162.2 eV (S 2p1/2) and 161.1 eV (S 2p3/2) in Figure 3c are characteristic of Ag₂S–CdS nanocomposites [44]. Thus, the XPS spectra confirmed the composition of AC11 [36].

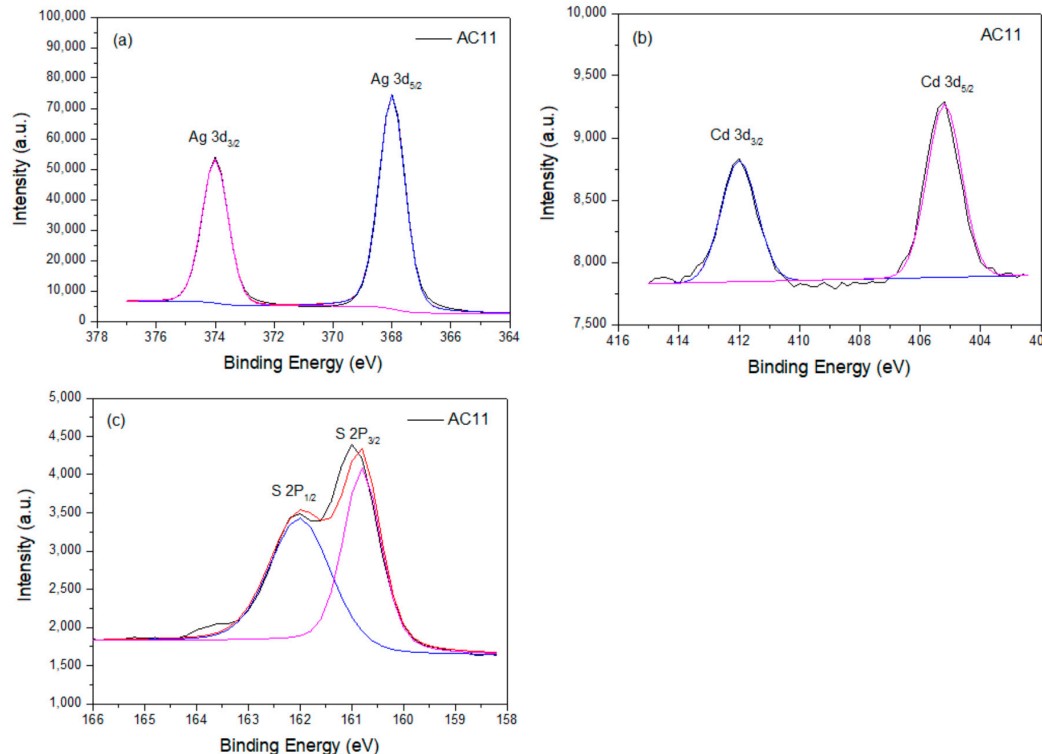

**Figure 3.** XPS spectra of the (**a**) Ag 3d, (**b**) Cd 3d, and (**c**) S 2p binding energies of AC11.

### 2.2. Characterization of PANI/Ag₂S–CdS Nanocomposites

The chemical structure of the PANI/Ag₂S–CdS nanocomposites was determined using FTIR spectroscopy. Figure 4 displays FTIR spectra of the PANI, PACI, and PACO nanocomposites. Weak absorptions for the NH stretching vibration appeared at 3500–

3750 cm⁻¹. The characteristic peaks near 1563 and 144 cm⁻¹ represented the stretching of the quinoid and benzenoid units, respectively, in the PANIs [5]. Moreover, absorption peaks at 1295 and 1234 cm⁻¹ represented C–N stretching vibrations with aromatic conjugation [5]. The vibration mode of the N = Q = N structure (Q: quinoid unit) appeared as a signal at a wavelength of 1110 cm⁻¹ [5]. The absorption peak for in-plane C–H bending on aromatic rings appeared at 1105 cm⁻¹ [5].

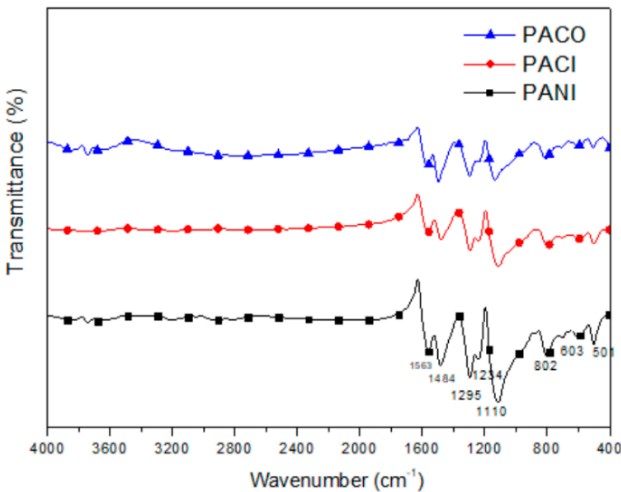

**Figure 4.** FTIR spectra of the PANI/Ag₂S–CdS nanocomposites.

Figure 5 presents TGA thermograms of PANI, PACI, and PACO. The thermal stabilities and char yield for PACI and PACO were greater than those of PANI, presumably because of the incorporation of AC11. The char yields of PACI and PACO were 5.1 and 12.3%, respectively, implying that AC11 was incorporated in the PACI and PACO samples at levels of 5.1 and 12.3 wt%, respectively, because PANI had degraded completely at 800 °C in the air. A higher content of AC11 existed in PACO than in PACI.

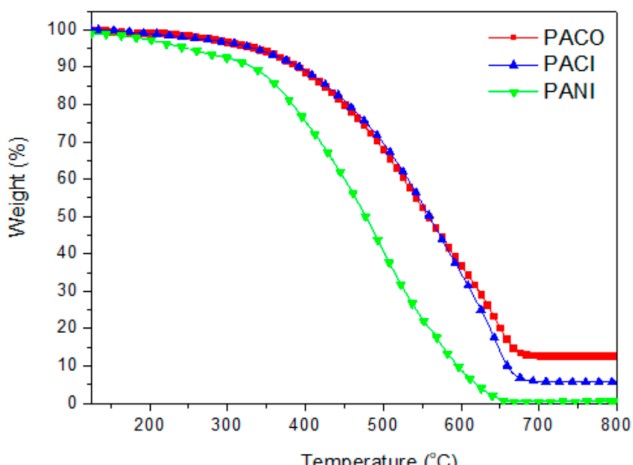

**Figure 5.** TGA thermograms of the PANI/Ag₂S–CdS nanocomposites.

Figure 6 displays the XPS spectra of PANI, PACI, and PACO. In the C 1s XPS spectrum of PANI (Figure 6a), the peaks located at 284.5, 284.8, and 285.7 eV can be correlated to the C = C, C–C, and C = N bonds, respectively. In Figure 6b, the N 1s spectrum of PANI features signals for quinoid phenyl (–NH=), benzenoid (–NH–), and quaternary ammonium (N⁺) units in the range 395.0–403.0 eV, consistent with those reported previously

[45]. Thus, the C 1s and N 1s XPS spectra confirmed the chemical structure of PANI. Moreover, the C 1s and N 1s XPS spectra of PACI and PACO were similar to those of PANI. In addition, the Ag 3d spectra of PACI (Figure 6c) and PACO (Figure 6d) featured peaks at 367.5 and 373.5 eV, which we attribute to Ag 3d$_{5/2}$ and Ag 3d$_{5/2}$, respectively. The intensities of the Ag 3d peaks for PACO were much higher than those for PACI, due to the low content of AC11 NPs embedded in the fibers of PACI, in contrast to the AC11 particles having been deposited on the surface of the PACO fibers; Khilar et al. reported that the inner energy levels of atoms can be affected by their chemical surroundings [46]. Furthermore, signals for Cd 3d$_{3/2}$ and Cd 3d$_{5/2}$ of negligible intensity were observed at 412.0 and 405.0 eV, respectively, in the XPS spectra of PACI and PACO. The intensities of the Cd 3d peaks were much lower than those of the Ag 3d peaks, consistent with the low content of CdS in AC11 (Figure 3a,b) [46]. Furthermore, S 2p peaks were observed in the range 158.0–164.0 eV for PACO (Figure 6e); they appeared with negligible intensity for PACI, consistent with its low content of AC11.

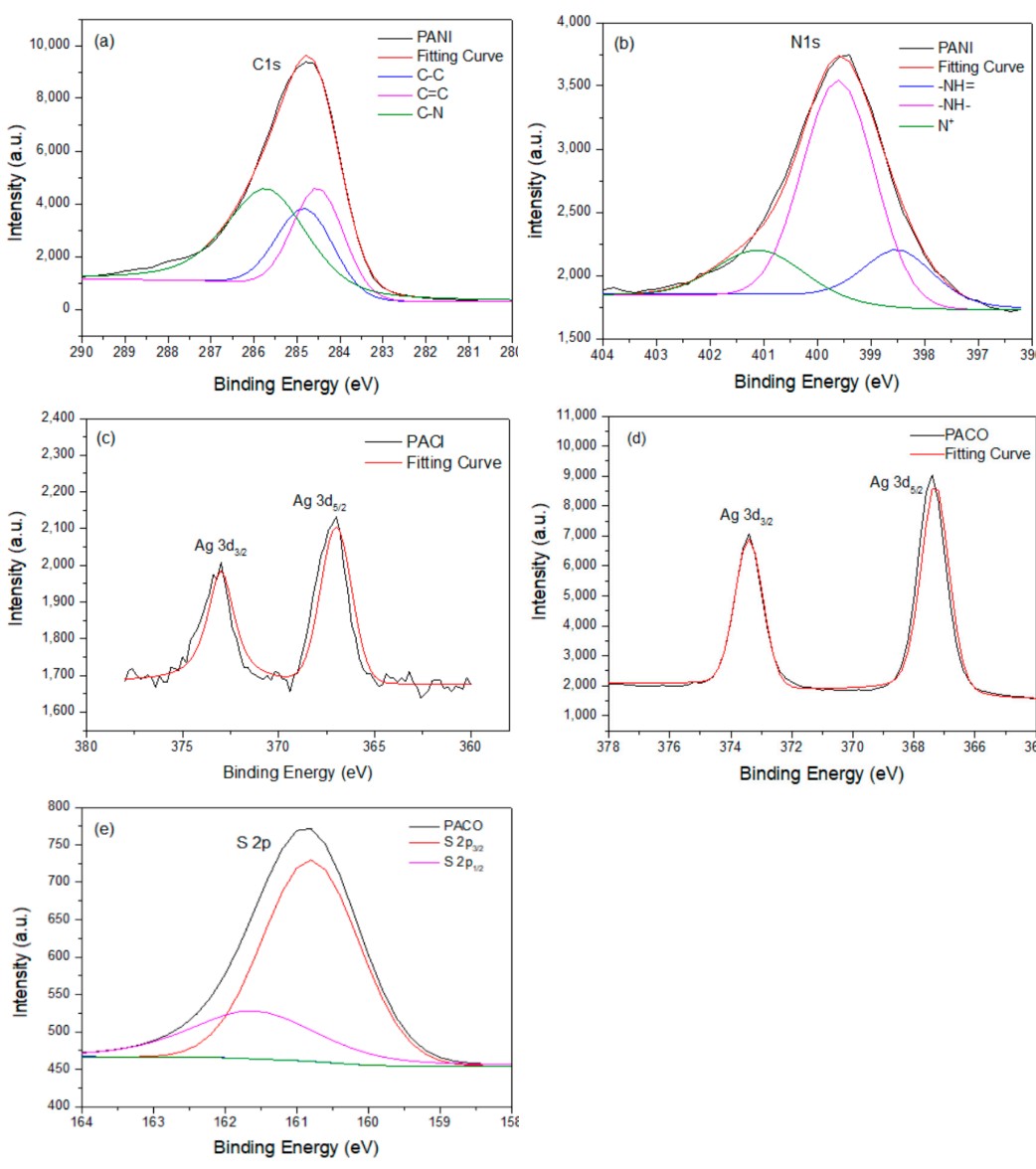

**Figure 6.** XPS spectra of (**a,b**) PANI, (**c**) PACI, and (**d,e**) PACO.

Figure 7 provides SEM images of PANI, PACI, and PACO. Figure 7a,b reveal that a porous network structure with many open pores and channels was created by the PANI fibers. The average diameter of the PANI fibers was approximately 30–50 nm. The average

length of the PANI fibers was in the range 200–300 nm. The average diameter of the fibers in PACI was slightly larger than that in PANI (Figure 7c,d). The diameters of the PACI and PACO fibers were approximately in the ranges 50–150 and 200–250 nm, respectively. Few AC11 NPs appeared on the surface of the PACI fibers; in contrast, a large amount of AC11 NPs had been deposited on the surface of PACO (Figure 7e,f). The numbers of open pores and channels in the conducting polymer-based network were relatively low. We used TEM to further analyze the morphologies of AC11, PANI, PACI, and PACO (Figure 8). The TEM images of AC11 and PANI in Figures 8a,b and 7b, respectively, reveal the aggregation of micro-scale particles in AC11. Relative to PANI, the average diameters of the particles were larger for PACI and, especially, for PACO (Figure 8c,d). AC11 had dispersed well on the nanoscale in the conducting polymer fibers for PACI, whereas aggregation of AC11 into large particle sizes was observed for PACO. The diameters of the AC11 particles in the PACI and PACO fibers were approximately in the ranges 1–5 and 10–80 nm, respectively (Figure 8c,d)].

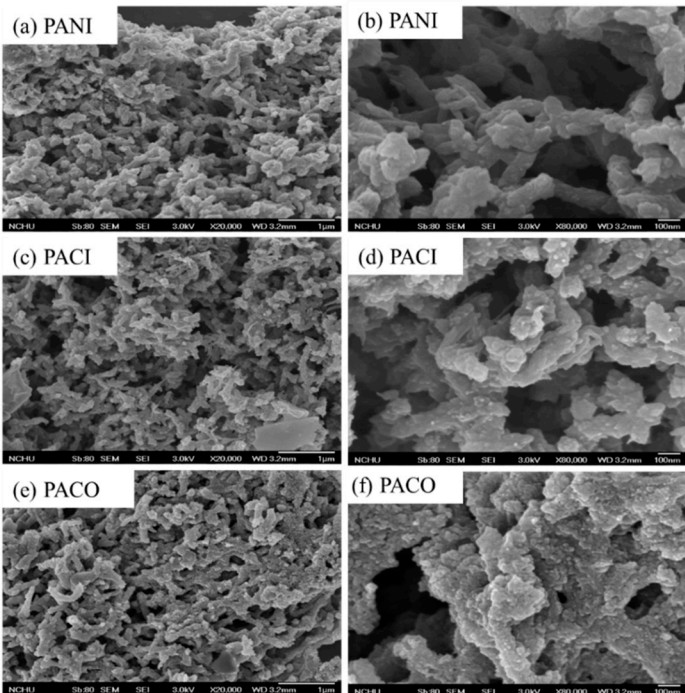

**Figure 7.** SEM images of (**a**,**b**) PANI, (**c**,**d**) PACI, and (**e**,**f**) PACO.

Figure 9 displays XRD patterns of the PANI/$Ag_2S$–CdS nanocomposites. The conducting polymer fibers of PANI had an amorphous structure, with a broad diffraction reflection at values of $2\theta$ from 10 to 40°. The patterns of the nanocomposites of PACI and PACO featured both the broad diffraction band of the conducting polymer fiber PANI and the diffraction peaks of AC11. The intensity of the AC11 diffraction peaks was higher for PACO than for PACI, due to the low content of AC11 embedded among the conducting polymer fibers for PACI and the large amount of AC11 coated and aggregated on the surface of the conducting polymer chains for PACO.

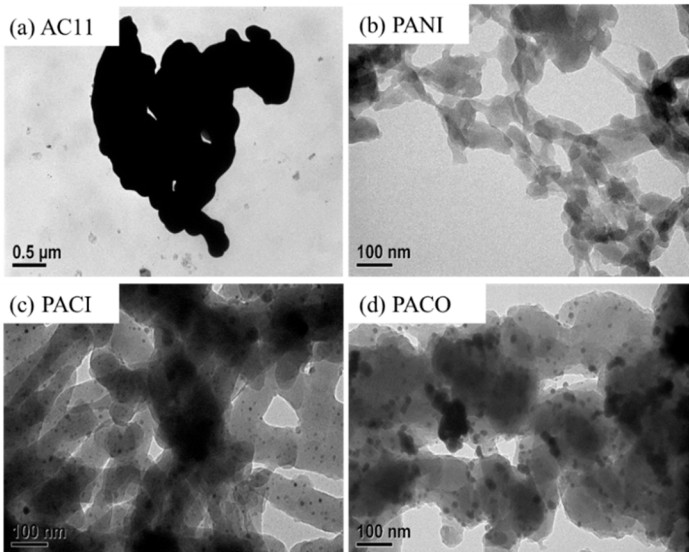

**Figure 8.** TEM images of (**a**) AC11, (**b**) PANI, (**c**) PACI, and (**d**) PACO.

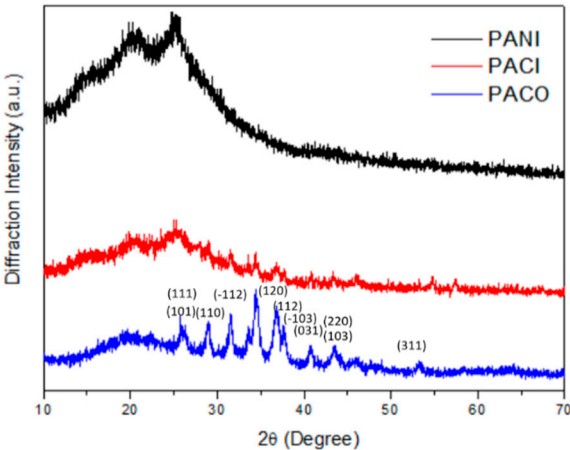

**Figure 9.** XRD patterns of PANI/Ag₂S–CdS nanocomposites.

To characterize the porous structures of the nanocomposites PANI, PACI, and PACO, we measured their $N_2$ adsorption/desorption behavior (Figure 10). All of the PANI/Ag₂S–CdS nanocomposites possessed porous structures dominated by mesopores, as suggested by the type-IV isotherms with distinct hysteresis loops appearing in their adsorption isotherms at moderate pressures [47,48]. The Brunauer–Emmett–Teller (BET) surface areas of PANI, PACI, and PACO were 58.68, 30.19, and 35.38 $m^2\ g^{-1}$, respectively. The surface area of PANI was higher than those of PACI and PACO because the fibers in PANI had much smaller average diameters than those in PACI and PACO. Moreover, PANI featured a large fraction of mesopores having sizes in the range from 10 to 100 nm, presumably because of the uniform distribution of its PANI fibers. The mesopore distribution curve of PACI was similar to that of PANI, although the pore volume of PACI was much smaller than that of PANI. Relative to PACI, PACO contained a higher fraction of smaller pores (<10 nm) as a result of the deposition of the Ag₂S-CdS nanostructures. Accordingly, the BET surface area of PACO was slightly larger than that of PACI.

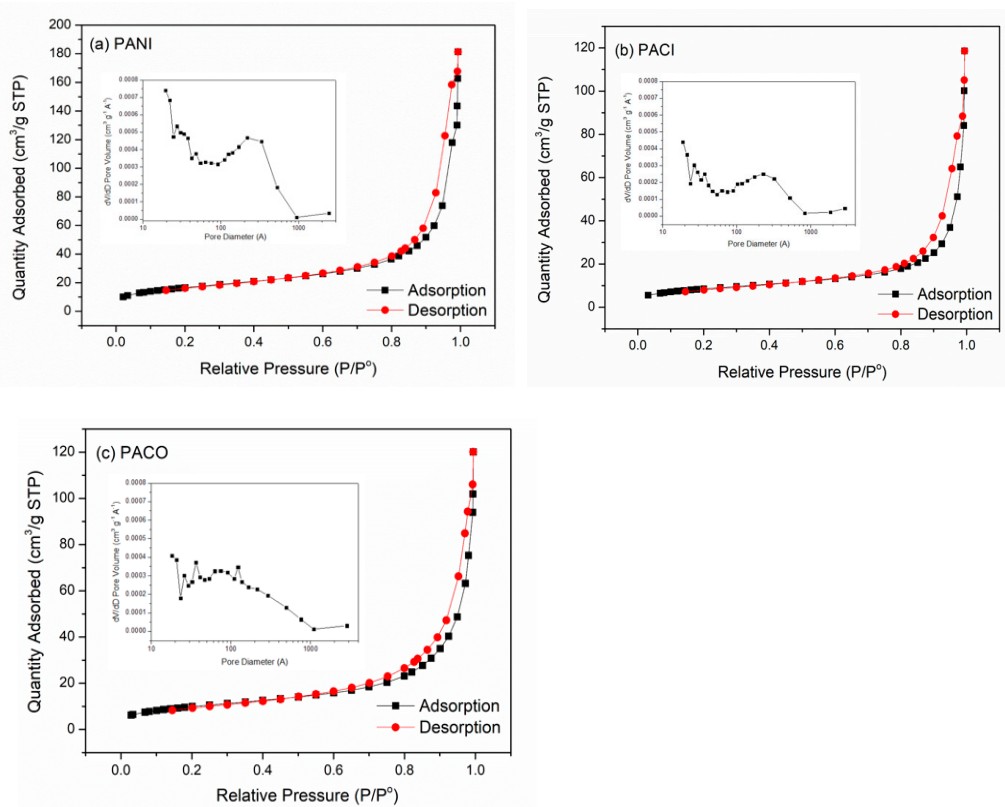

**Figure 10.** $N_2$ sorption isotherms of PANI, PACI, and PACO; insets: corresponding pore size distributions.

### 2.3. Morphologies and Electrochemical and PV Properties of PANI/Ag2S–CdS Composite-Based Counter Electrodes

We spin-coated the PANI/Ag2S–CdS nanocomposites (PANI, PACI, PACO) onto the surfaces of ITO-deposited glass substrates and tested them for their potential use as the counter electrodes of DSSCs. Figure 11 displays SEM images of the PANI-S, PACI-S, and PACO-S electrodes. The morphologies of these counter electrodes were similar to those of the PANI/Ag2S–CdS nanocomposites. The PANI-S and PACI-S electrodes possessed porous network structures with many open pores and channels. Thus, we expected the PANI-S and PACI-S electrodes to favor the absorption of the electrolyte in their PANI/Ag2S–CdS composite films and, thereby, enhance the electrocatalytic activity for the $I_3^-/I^-$ redox reaction [49]. In contrast, the PACO-S electrode featured a lower content of pores, consistent with a larger amount of AC11 having been deposited on the surface of its conducting polymer fibers.

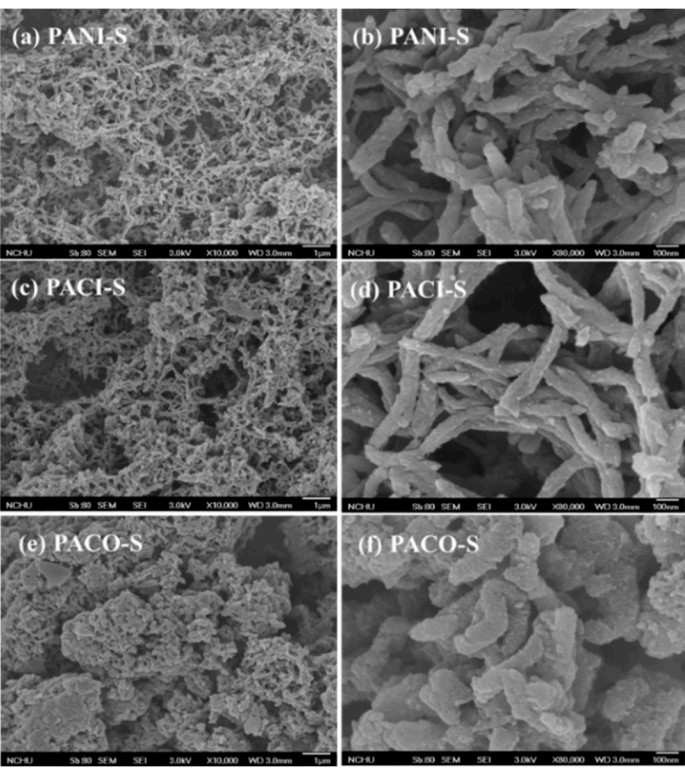

**Figure 11.** SEM images of the counter electrodes (**a,b**) PANI-S, (**c,d**) PACI-S, and (**e,f**) PACO-S.

Figure 12 presents cyclic voltammograms of the PANI-S, PACI-S, and PACO-S counter electrodes. Each of the electrodes provided two pairs of redox waves; we assign the left and right pairs to the oxidation and reduction of $I^-/I_3^-$ and $I_2/I_3^-$, respectively. The PANI/AC11-based electrodes served to catalyze the reduction of $I_3^-$ to $I^-$; thus, our research focus was primarily associated with the characteristics of the left-hand pair of peaks. The measured peak current density and overpotential losses were used to evaluate the electrocatalytic activities of the PANI/AC11-based electrodes [50]. The redox peak for the PACI-S electrode was shifted slightly toward a negative potential, relative to that for the PANI-S electrode, implying the presence of a larger overpotential loss in the former [50]. Nevertheless, the peak current density of the PACI-S electrode was much higher than that of the PANI-S electrode. The electrocatalytic activity of the PACI-S electrode was greater than that of the PANI-S electrode. SEM (Figure 7d) and TEM (Figure 8c) images indicated that the AC11 particles were distributed uniformly in the PANI fibers. A synergistic effect of the PANI and AC11 components resulted in the higher electrocatalytic activity of PACI-S [21,30]. In contrast, the catalytic activity of the PACO-S electrode was much lower than that of PANI-S electrode. For the PACO-S electrode, a large amount of AC11 had been deposited on the surface of the PANI fibers (Figures 7f and 8d), resulting in poor contact between the PANI and the liquid electrolyte as well as suppression of the synergistic effect of the PANI and AC11 components. Therefore, at higher peak current densities, the electrocatalytic activities of the PACI-S and PANI-S electrodes were much greater than that of the PACO-S electrode. Thus, CV suggested that, among our tested samples, the PACI-S electrode would be the most suitable counter electrode for DSSCs.

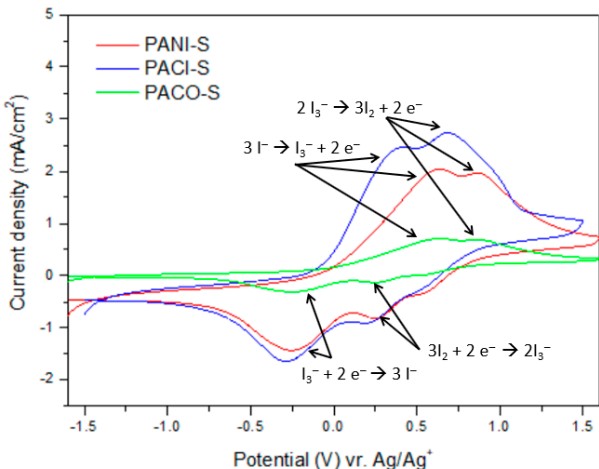

**Figure 12.** Cyclic voltammograms of PANI-S, PACI-S, and PACO-S electrodes (scanning rate: 100 mV s$^{-1}$; electrolyte: 1 mM I$_2$, 10 mM LiI, and 0.1 M LiClO$_4$ in MeCN).

We used EIS to measure the electrocatalytic activities of the PANI/Ag$_2$S–CdS nanocomposite-based counter electrodes. Figure 13 presents Nyquist plots of symmetric cells incorporating the PANI-S, PACI-S, and PACO-S counter electrodes [9]. The equivalent circuit model in the inset to Figure 13 was used for curve-fitting of the electrochemical impendence of the cells. The charge-transfer resistance ($R_{ct}$) of each counter electrode was taken as half the value of the real semicircles in the middle-frequency range (10$^1$–10$^5$ Hz) [9,39]. The values of $R_{ct}$ of the cells featuring the PANI-S, PACI-S, and PACO-S counter electrodes were 158.1, 102.3, and 603.5 Ω cm$^2$, respectively. The value of $R_{ct}$ of the PACI-S electrode-based cell was lower than that of the PANI-S electrode-based cell, consistent with the higher electrocatalytic activity of the PACI-S electrode. In addition, the value of $R_{ct}$ of the PACO-S electrode-based cell was much higher than that of the PANI-S electrode-based cell. These findings are consistent with the CV behavior of the PANI-S, PACI-S, and PACO-S electrodes. The synergistic effect of PANI and AC11 led to the lower value of $R_{ct}$ of the PACI-coated counter electrode.

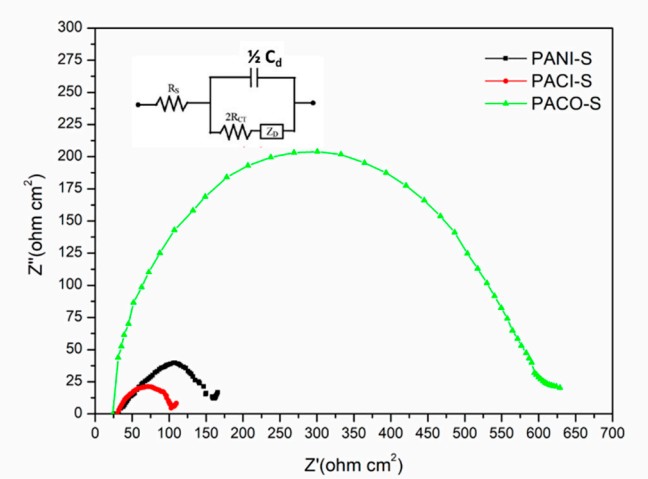

**Figure 13.** Nyquist plots of electrochemical cells containing PANI-S, PACI-S, and PACO-Scounter electrodes (inset: equivalent circuit).

Figure 14 displays the best PV performances of the DSSCs incorporating the PANI-S, PACI-S, and PACO-S counter electrodes; Table 1 summarizes the statistical values of their open-circuit voltages ($V_{OC}$), short-circuit current densities ($J_{SC}$), fill factors (FFs), and PCEs. Four runs of PV evaluation tests were performed for each DSSC sample. The values

of $V_{OC}$ and the FFs of the DSSCs containing the PANI-S and PACI-S electrodes (DSSC I and DSSC II, respectively) were almost identical, whereas the values of $J_{SC}$ and the PCEs of DSSC II were much larger than those of DSSC I. We attribute this behavior to the higher electrocatalytic activity and lower value of $R_{ct}$ of the PACI-S electrode in comparison with those of the PANI-S electrode. In addition, the PV parameters of PACO-S-based DSSC III were much lower than those of the PANI-S- and PACI-S-based DSSC-I and DSSC II, respectively, presumably because the PACO-S electrode exhibited lower electrocatalytic activity and a larger value of $R_{ct}$ than those of the PANI-S and PACI-S electrodes. We conclude that the PV properties of the PANI/Ag$_2$S–CdS nanocomposite-based counter electrodes were strongly dependent on the distribution and condition of the Ag$_2$S–CdS NPs in the PANI fibers.

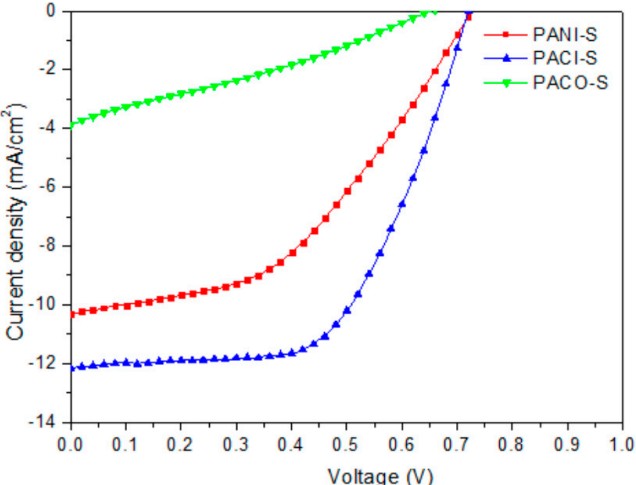

**Figure 14.** Current density–voltage plots of DSSCs featuring PANI-S, PACI-S, and PACO-S counter electrodes (light intensity: 100 mW cm$^{-2}$).

**Table 1.** PV performance of DSSCs featuring PANI-S, PACI-S, and PACO-S counter electrodes (measured under AM 1.5G illumination, 100 mW cm$^{-2}$).

| DSSC | Counter Electrode | $V_{OC}$ (V) | $J_{SC}$ (mA cm$^{-2}$) | FF | PCE (%) | Best PCE (%) |
|---|---|---|---|---|---|---|
| DSSC I | PANI-S | 0.71 ± 0.01 | 10.1 ± 0.02 | 0.54 ± 0.01 | 3.87 ± 0.14 | 4.01 |
| DSSC II | PACI-S | 0.71 ± 0.01 | 12.05 ± 0.01 | 0.57 ± 0.01 | 4.87 ± 0.17 | 5.04 |
| DSSC III | PACO-S | 0.64 ± 0.02 | 4.00 ± 0.02 | 0.27 ± 0.02 | 0.69 ± 0.08 | 0.77 |

### 3. Materials and Methods

*3.1. Chemicals*

Silver nitrate, cadmium acetate, aniline, oxidant ammonium peroxydisulfate (APS), thiourea, MeOH, EtOH, and *N*-methyl-2-pyrrolidone (NMP) were purchased from Sigma–Aldrich (St. Louis, MO, USA), Acros (Fukuoka, Japan), and TCI Chemical (Tokyo, Japan), and used without purification.

*3.2. Ag$_2$S–CdS Nanocomposite*

**CdS NPs**

Sodium dodecyl sulfate (SDS, 0.05 mol) was dissolved in DI water (100 mL). Cadmium acetate (0.01 mol) and thiourea (0.01 mol) were added into the SDS solution and stirred at 60 °C for 12 h. The mixture was transferred to a Teflon-lined stainless-steel autoclave, maintained at 200 °C for 4 h, and then cooled to room temperature. The powder

was washed several times with EtOH and DI water. The product CdS NPs were dried in a vacuum oven for 24 h [36,51].

**Ag$_2$S**

Silver acetate (0.01 mol) and thiourea (0.01 mol) were stirred in DI water (200 mL) under reflux at 80 °C for 4 h. Hydrazine monohydrate (10 mL) was added and then the mixture was heated under reflux at 80 °C for 5 h. The solution was cooled to room temperature. The precipitate was washed several times with EtOH and DI water. The product Ag$_2$S was dried in a vacuum oven for 24 h.

**Ag$_2$S–CdS nanocomposite AC11**

Silver nitrate (0.01 mol) and cadmium acetate (0.01 mol) were dissolved in DI water (100 mL). Thiourea (50 mL) was added dropwise into the mixture, which was then stirred at 80 °C for 12 h. SDS (10 wt%) was added and then the mixture was stirred at 60 °C for 3 h. The black precipitate was collected through centrifugation and washed several times with distilled water, EtOH, and acetone. AC11 was obtained after drying in a vacuum oven for 24 h.

*3.3. PANI/Ag$_2$S–CdS Nanocomposites*

**PANI**

Solution A was prepared as a 1 M HCl (100 mL) solution containing aniline (1.86 g, 0.0200 mol); solution B as a 1 M HCl (100 mL) solution containing APS (4.56 g, 0.0200 mol). Solution B was added dropwise to solution A as it was being vigorously stirred at 0 °C under a N$_2$ atmosphere. The mixture was then stirred for 6 h. The crude product was washed several times with distilled water and MeOH. PANI (1.35 g) was obtained after drying in a vacuum oven for 24 h.

**PACI**

Solution A was prepared as a 1 M HCl (100 mL) solution containing AC11 powder (250 mg) and aniline (1.86 g, 0.0200 mol); solution B as a 1 M HCl (100 mL) solution containing APS (4.56 g, 0.0200 mol). Solution A was stirred at RT for 20 min and then solution B was added dropwise with vigorous stirring at 0 °C under a N$_2$ atmosphere. The mixture was stirred for 6 h. The crude product was washed several times with distilled water and MeOH. PACI (1.52 g) was obtained after drying in a vacuum oven for 24 h.

**PACO**

A mixture of PANI (0.52 g), silver nitrate (0.20 g, 1.2 mmol), and cadmium acetate (0.32 g, 1.2 mmol) in NMP (100 mL) was stirred for 30 min. Thiourea (0.22 g, 3.6 mmol) was added and then the mixture was heated at 60 °C for 8 h. The black precipitate was washed several times with distilled water and MeOH. PACO (0.72 g) was obtained after drying in a vacuum oven for 24 h.

*3.4. Counter Electrodes*

3.4.1. PANI-Based Electrodes (PANI-S)

An ITO-coated glass substrate was washed well and then cleaned through O$_2$ plasma treatment. A solution of PANI (16 mg/mL) in EtOH was subjected to ultrasonication for 20 min to form an even suspension. The PANI solution was spin-coated (1000 rpm, 60 s) onto the pretreated ITO glass and then dried by heating at 100 °C for 10 min.

3.4.2. PACI-Based Electrodes (PACI-S)

PACI-S electrodes were prepared using a fabrication process similar to that for PANI-S.

3.4.3. PACO-Based Electrodes (PACO-S)

PACO-S electrodes were prepared using a fabrication process similar to that for PANI-S.

*3.5. DSSCs*

DSSCs were prepared with an active area of 0.25 cm$^2$, according to a previously reported method [5]. First, the photoelectrode was prepared by coating a TiO$_2$ thin film onto an FTO-deposited glass substrate (sheet resistance: 15 Ω sq$^{-1}$; Solaronix). The TiO$_2$ paste was prepared through mechanical grinding of a mixture of TiO$_2$ (1 g), aqueous acetic acid (5 wt%, 5 mL), and polyoxyethylene (10) octylphenyl ether (Triton X-100, 0.4 mL). The prepared TiO$_2$ paste was coated onto the FTO-deposited glass using the doctor blade method [8,9]. After coating, the TiO$_2$ electrode was dried at 100 °C and then sintered at 500 °C for 2 h. The TiO$_2$ film had a thickness of approximately 10 m and was used as the photo-anode. The surface of the TiO$_2$-based photoanode was soaked in a 0.5 mM solution of ruthenium dye (N719 dye, Solaronix) in MeCN/*tert*-butanol. The amount of N719 dye anchored on the TiO$_2$-based photoanode was 4.18 μmol g$^{-1}$ [8,9]. The PANI/AC11 nanocomposite was coated on the ITO glass to function as a counter electrode. The liquid electrolyte was prepared by mixing I$_2$ (0.05 M), LiI (0.1 M), 4-*tert*-butylpyridine (0.2 M), 1-propyl-2,3-dimethylimidazolium iodide (0.6 M), and GuSCN (0.1 M) in MeCN (4 mL). Finally, the N719 dye–adsorbed photoelectrode and the PANI/AC11-based counter electrode were placed side by side and the electrolyte was poured into the space between them. After encapsulation, the solar cell was obtained.

*3.6. Characterization*

The chemical structures of the PANI/AC11 nanocomposites were characterized using Fourier transform infrared spectroscopy (HORIBA FT-720 FTIR spectrometer, HORIBA Inc., Tainan City, Taiwan). Thermogravimetric analysis (TGA) was performed in air using a thermogravimetric analyzer (TA Instruments, TGA-2050, New Castle, DE, USA) operated at a heating rate of 10 °C min$^{-1}$. X-ray photoelectron spectroscopy (XPS; ESCALAB 250Xi, Thermo Fisher, Massachusetts, USA) was performed to study the surface chemical components of the samples. X-ray diffractometry (XRD, Rigaku RINT 2000, Tokyo, Japan) of the PANI/AC11 samples was performed using a Rigaku diffractometer (Rigaku RINT 2000, Tokyo, Japan) and Ni-filtered Cu Ka radiation. The redox potentials of films of PANI and the PANI/AC11 nanocomposites were determined through CV, using a BAS 100B electrochemical analyzer operated at a scanning rate of 100 mV s$^{-1}$; a solution of I$_2$ (0.001 M), LiI (0.01 M), and LiClO$_4$ (0.1 M) in deoxygenated dry MeCN was used as the electrolyte; the working electrode was a PANI or PANI/AC11 nanocomposite film; the reference electrode was saturated non-aqueous Ag/AgCl. The nanostructures of the prepared PANI and PANI/AC11 nanocomposite counter electrodes were analyzed using cold field emission scanning electron microscopy (FESEM; Hitachis-4800, Taipei, Taiwan) and high-resolution transmission electron microscopy (HRTEM; JEOL JEM-2010, Tokyo, Japan). For the TEM investigation, films of PANI and the PANI/AC11 nanocomposite were delaminated from the ITO-coated substrate in an ultrasonicator. The PANI and PANI/AC11 films were floated on water and then placed on a 200-mesh copper TEM grid (Agar Sci.). Electrochemical impedance spectroscopy (EIS) was used to study the catalytic activities of the PANI/AC11-based counter electrodes. The EIS measurements of the DSSCs were performed using a CHI6273D electrochemical work station (CH Instruments, Inc., Austin, TX, USA) over the frequency range from 0.01 Hz to 1 MHz with two identical counter electrodes (working area: 1 cm$^2$) in a symmetrical cell [9,52,53]. The electrolyte was the same as that used for the DSSC samples. The two counter electrodes of the cell were separated by a Surlyn spacer film having a thickness of 60 μm. The PV parameters of the DSSCs were measured using a programmable electrometer equipped with current and voltage sources (Keithley 2400) under solar light illumination (100 mW cm$^{-2}$) from an AM1.5 solar simulator (Newport Oriel 96000).

## 4. Conclusions

We have developed PANI/Ag₂S–CdS nanocomposites (PANI, PACI, PACO) as efficient electrocatalysts for triiodide reduction in DSSCs. The PANI- and PACI-based counter electrodes (PANI-S and PACI-S, respectively) possessed porous network structures with many open pores and channels—a favorable arrangement for absorption of the electrolyte through trapping of the liquid in their PANI/AC11 composite films. As a result, the counter electrodes displayed high electrocatalytic activities for the I₃⁻/I⁻ redox reactions. Relative to the PANI-S electrode, synergistic effects of the PANI and AC11 components resulted in higher catalytic activity for the PACI-S electrode; as a result, we observed lower charge-transfer resistance for the PACI-S-based counter electrode. The PV performance of the PACI-S-based DSSC was much better than that of the PANI-S-based DSSC. In contrast, the PACO-S-based DSSC displayed low catalytic activity, high charge-transfer resistance, and poor PV performance as a result of aggregation of the AC11 NPs on the surface of the PANI fibers. We conclude that the catalytic activity of the PANI/Ag₂S–CdS nanocomposites was strongly dependent on (i) the size and distribution of the AC11 NPs in the PANI fiber and (ii) the microporous structure of the composite layer.

**Author Contributions:** Conceptualization R.-H.L.; methodology, M.K.; validation, R.-H.L.; investigation, K.K., T.-C.C., H.-K.Y. ; resources, R.-H.L.; writing—original draft preparation, R.-H.L.; writing—review and editing, T.-L.W., T.-H.W., C.-C.K., R.-H.L.; supervision, R.-H.L.; project administration, R.-H.L.; funding acquisition, R.-H.L. All authors have read and agreed to the published version of the manuscript.

**Funding:** This research was funded by the Ministry of Science and Technology (MOST) of Taiwan (grant no. MOST 109-2221-E-005 -070-MY3.

**Data Availability Statement:** Data is contained within the article.

**Acknowledgments:** The authors are grateful for the financial support provided by the MOST of Taiwan.

**Conflicts of Interest:** The authors declare no conflict of interest.

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
