# Peer review of "Polyaniline/Ag2S–CdS Nanocomposites as Efficient Electrocatalysts for Triiodide Reduction in Dye-Sensitized Solar Cells"

_catalysts, doi:10.3390/catal11040507_

Round 1
Reviewer 1 Report
This manuscript reports the study of the performance of three different electrocatalysts based on Ag2S-CdS nanocomposites. The authors carried out the electrocatalysts characterization by SEM-TEM, XRD, FTIR and, then, through electrochemical measurements. Before the publication on Catalysts journal, the authors need to address some issues based on the following comments:
1) Please, remove the editor indications at the beginning of Introduction section;
2) How do you explain that nanoparticles of AC11 are smaller than those of Ag2S and CdS?
3) Apart the peak intensity recorded during CV measurements, do you also consider the overpotential values? That related to PANI-S seems the lowest (Fig. 10);
4) Line 348: “The charge-transfer resistance (Rct) of the counter electrode was taken as half the value of the real semicircles on the high-frequency side” This is not correct, Rct can be estimated from the low frequency impedance. Please, verify the bibliographic reference and, consequently, the text of the manuscript.
Reviewer 2 Report
The article format looks not obeyed to the journal templates. I recommend author to re-submit the manuscript.
Figure 11 looks slugish. The author should show the better data and fitting line.
Introduction: delete following "The introduction should briefly place the study in a broad context and highlight why 32 it is important...... See the end of the docu- 39 ment for further details on references."
Replace line 230: "m" to "micro m"
Reviewer 3 Report
The manuscript "Polyaniline/Ag2S-CdS nanocomposites as efficient electrocatalyst for triiodide reduction in dye-sensitized solar cell" by Kuo et al. describes efficient electrocatalysts for the reduction of tri-iodide. The electrochemical properties of the nanocomposite were well investigated. but the charecterization is insufficient performed or explained.
In detail:
p.5 .. the average diamter of the AgS particle ranged from 0.1 to 0.5 m (Figure 1(b); I am missing the unit; please specify the size of nanoparticles escpecially on the nanocomposites wiht the PACI and PACO.
p.7 What is the scientific output ot the XPS investigations. I am missing the evaluation and some investigations of the nanocomposites with the PANI ( PACI and PACO). Furthermore, a deconvolution of the peaks with an exact determination of the binding energy and a quantification are missing. A minor remark: binding energy axis is usually in the other direction (from high to low binding energies)
p. 8 and p.10: Please show the pores in the TEM or SEM. I am missing any quantitative statements about the pores or the surface area of the sample. BET is an appropriate methods to obtain this information.
Some minor remarks.
abstract: please mention the S source in the abstract or remove the Ag and Cd source.
introduction: please remove the first sentences (until line 40). They are the explanation from MDPI.
p. 9 l. 299: not "diffraction band", diffraction reflexes or reflections, at values of 2 , there is something missing, probably Θ
Round 2
Reviewer 3 Report
The authors did a good job to improve the quality of the manuscript. Only one small remark: in Fig. 6e) the ratio between p 3/2 and p 1/2 is not reasonable, it should be 2:1. This should be changed.
